# Injection Vaccines Formulated with Nucleotide, Liposomal or Mineral Oil Adjuvants Induce Distinct Differences in Immunogenicity in Rainbow Trout

**DOI:** 10.3390/vaccines8010103

**Published:** 2020-02-25

**Authors:** Kasper Rømer Villumsen, Per Walter Kania, Dennis Christensen, Erling Olaf Koppang, Anders Miki Bojesen

**Affiliations:** 1Department of Veterinary and Animal Sciences, Faculty of Health and Medical Sciences, University of Copenhagen, DK-1870 Frederiksberg, Denmark; pwk@sund.ku.dk; 2Adjuvant Research, Statens Serum Institut, DK-2300 Copenhagen S, Denmark; den@ssi.dk; 3Section of Anatomy and Pathology, Institute of Basal Sciences and Aquatic Medicine, Faculty of Veterinary Medicine, Norwegian University of Life Sciences, N-0454 Oslo, Norway; erling.o.koppang@nmbu.no

**Keywords:** adjuvant, rainbow trout, adverse effects, CAF01, CpG ODN, mineral oil

## Abstract

Protection facilitated by the widespread use of mineral oil adjuvanted injection vaccines in salmonid fish comes with adverse effects of varying severity. In this study, we characterized the immunological profiles of two alternative vaccine formulations, both with proven efficacy and an improved safety profile in rainbow trout. Experimental injection vaccines were prepared on an identical whole-cell *Aeromonas salmonicida* bacterin platform and were formulated with CpG oligodeoxynucleotides, a liposomal (CAF01) or a benchmark mineral oil adjuvant, respectively. A naïve group, as well as bacterin and saline-injected groups were also included. Following administration, antigen-specific serum antibody titers, the tissue distribution of immune cell markers, and the expression of immune-relevant genes following the in vitro antigenic restimulation of anterior kidney leukocytes was investigated. Immunohistochemical staining suggested prolonged antigen presentation for the particulate formulations and increased mucosal presence of antigen-presenting cells in all immunized fish. Unlike the other immunized groups, the CAF01 group only displayed a transient elevation in specific antibody titers and immunohistochemical observations, and the transcription data suggest an increased role of cell-mediated immunity for this group. Finally, the transcription profile of the CpG formulation approached that of a T_H1_ profile. When compared to the benchmark formulation, CAF01 and CpG adjuvants induce slight, but distinct differences in the resulting protective immune responses. This is important, as it allows a broader immunological approach for the future development of safer vaccines.

## 1. Introduction

Successful prophylactic strategies in the form of vaccination programs represent a fundamental factor in the success of modern aquaculture. Serving to both minimize the use of antibiotics, as well as to facilitate increased production [1], vaccines are administered to large numbers of farmed fish, including salmonids such as Atlantic salmon (*Salmo salar*) and rainbow trout (*Oncorhynchus mykiss*), on an annual basis. While vaccination against some bacterial pathogens has been successfully performed by bath exposure [2,3,4,5], others require the injection of adjuvanted vaccine formulations in order to induce a satisfactory level of protection [6]. Efficacious vaccination against *Aeromonas salmonicida* subspecies *salmonicida*, the Gram-negative bacterial etiological agent of furunculosis in a wide range of fish species [7,8], has been found to require the intraperitoneal delivery of inactivated bacteria formulated with adjuvants, typically mineral oil [9,10,11]. A number of studies, however, have demonstrated adverse effects of varying severity following the intraperitoneal injection of mineral oil adjuvanted vaccines in salmon and rainbow trout [11,12,13,14,15,16,17,18,19]. These findings warrant a continued effort to identify potential alternative vaccine formulations that can offer a similar or better protection, while at the same time reducing or preferably eliminating adverse effects.

We have previously investigated the protective and adverse effects induced by formulating a whole-cell bacterin of inactivated *A. salmonicida* with either CpG oligodeoxynucleotides (CpG ODNs) or the cationic adjuvant formulation 01 (CAF01) liposomal adjuvant compared to a benchmark formulation of the same bacterin with Freund’s incomplete adjuvant (FIA) [20]. The choice of adjuvants was based on expected safety profiles, as well as the suspected differential induction of immune responses. Reductions in both microscopic and macroscopic pathological changes were observed for both the CpG ODN and CAF01 formulations when compared to the mineral oil benchmark formulation. Meanwhile, an experimental infection revealed no differences in the induced protection between the three vaccine formulations. As our earlier work thus demonstrated induced protection as well as varying safety profiles conferred by three, theoretically distinctive experimental vaccines, as well as by the injection of bacterin alone [20], further investigation of the underlying immune responses for each vaccine formulation was warranted.

The immunogenic effects of CpG ODNs have been reported to be determined by composition, length, and the presence of certain, species-specific sequence motifs [21]. In teleosts, unmethylated CpG ODNs have been shown to act through activation of Toll-like receptor 9 in Atlantic salmon [22], and specific CpG ODNs have been shown to induce a proliferation of anterior kidney, spleen, and peripheral blood leukocytes in rainbow trout, as well as IFN-ϒ from Atlantic salmon anterior kidney cells [21,23,24]. The immunological effects of the CAF01 adjuvant have not been addressed in a teleost model prior to this study. In murine models, however, experimental subunit vaccines formulated with the liposomal adjuvant CAF01 have been shown to promote IFN-ϒ and IL-17 responses and an IgG2-skewed antibody profile [25,26].

The aim of the present study was to examine the immune responses following the intraperitoneal administration of these vaccines in rainbow trout, with regard to both humoral and cell mediated immunity. Individual groups of rainbow trout were vaccinated with either full vaccine formulations, including inactivated bacterin and either CpG ODNs, CAF01, or FIA, respectively, or the bacterin without an adjuvant. In addition, two control groups were included: one injected with tris-buffered saline (TBS) and one left as a naïve control group. Following vaccination, the development of *A. salmonicida*-specific immunoglobulins was followed by enzyme-linked immunosorbent assay (ELISA). The tissue distribution of major histocompatibility complex class II- (MHCII), CD8-, IgM, and IgT-positive cells was investigated by immunohistochemical (IHC) staining. Finally, to compare the T-cell responses of each experimental group to known mammalian and teleost T-helper (T_H_) cell profiles, the in vitro restimulation of leukocytes isolated from the anterior kidney of fish from each group with *A. salmonicida* was performed. Leukocyte responses were then investigated using real-time quantitative polymerase chain reaction (RT-qPCR) to assess the transcription level of a panel of transcription factors, cytokines, complement factors, and cell-surface markers.

## 2. Materials and Methods

Disclaimer: The work presented in this research article is part of a larger study, parts of which have previously been published [20]. Since the present study builds upon the same experimental setup, experimental animals, vaccine formulations, immunizations, and sampling program, those details have already been described and published in the abovementioned publication. Certain details concerning the experimental setup are repeated here to allow a complete presentation of the present study. The results presented here, however, have not been published prior to this publication.

### 2.1. Ethics Statement

The Danish Animal Experimentation Inspectorate approved all experimental animal protocols performed in this study, including handling and treatment of the animals, under the Danish law on the use of experimental animals. The study was performed under license no. 2014-15-0201-00252.

### 2.2. Fish

As previously described [20], rainbow trout eggs (all-female, AquaSearch Fresh, Fousing strain, AquaSearch ova ApS, Denmark) were disinfected and hatched at AquaBaltic, Nexø, Denmark. Once hatched, the fish were moved to recirculated 500 L fiberglass tanks for rearing (20 L/min, 15 °C, 5 h/19 h light/dark cycle) and fed commercial, pelleted feed (BioMar, Denmark, 0.5% of total fish weight/tank/day).

### 2.3. Vaccines

As described elsewhere [20], three experimental vaccines were formulated based on the same whole-cell bacterin, and one of three different adjuvants. *Aeromonas salmonicida* subsp. *salmonicida* strain 040617-1/1A (kindly provided by Dr. Inger Dalsgaard, DTU, Denmark), was cultured, validated, inactivated, and prepared as described previously [20]. The final CFU/mL was adjusted to 2 × 10^9^ CFU/mL in TBS.

Three experimental vaccines were then formulated by supplementing the inactivated bacterin with one of the following adjuvants:(I)CpG Oligodeoxynucleotides: Nucleotides were synthesized with a phosphorothioate backbone and a 40% GC-content (Eurofins Genomics, Germany). Both sequence and dosage were based on that of CpG ODN 2143 published by Carrington and Secombes [27]:5′-TTCGTCGTTTTGTCGTTTTGTCGTT-3′The CpG ODNs were purified by high-performance liquid chromatography and lyophilized prior to shipment. Prior to use, they were reconstituted in TBS according to the manufacturer’s instructions.(II)CAF01 (Statens Serum Institut, Denmark): Liposomal formulation of dimethyldioctadecyl-ammonium bromide and α,α-trehalose 6,6-dibehenate (5:1) suspended in TBS.(III)FIA (Sigma-Aldrich, USA): A mixture of 85% paraffin oil and 15% mannide monooleate by volume. Forms a water-in-oil (W/O) emulsion when combined with an antigen in an aqueous solution.

### 2.4. Vaccination and Experimental Group Setup

In keeping with the guidelines for the currently available furunculosis vaccine in Danish aquaculture, the fish were vaccinated upon reaching an average weight of >15 g [28]. As previously described [20], the fish were injected and/or allocated to one of six experimental groups as shown in Table 1.

A total of 240 individuals were allocated to each group. Immediately prior to injection, the fish were anesthetized in a separate tank, holding 100 mg/L tricaine methanesulfate (MS-222, Sigma-Aldrich, USA) in tank water. All injections were performed according to commercial guidelines [29] using a heat-sterilized, fixed volume Kaycee FishJector vaccine pistol (Kaycee Veterinary Products Ltd., UK). The vaccine pistol was rinsed by multiple ejections of TBS between each change of experimental formulation. Prior to use, each experimental formulation was thoroughly mixed, and the emulsified state of the FIA formulation was confirmed. Immediately following injection, the fish of each group were transferred to clean water in their respective 500 L fiberglass holding tanks.

### 2.5. Sampling

Prior to all sampling, fish were euthanized in a separate tank with 200 mg/L MS-222 in tank water. Blood samples were collected by caudal vein puncture using a 25G needle both prior to vaccination (10 fish), as well as 3, 6, and 10 weeks post vaccination (10 fish/group). The sampling time points correspond to 315, 630, and 1005 degree-days post vaccination. Degree days, a measure of time that enables comparison between the immunological kinetics of fish kept at differing temperatures, are calculated as number of days multiplied by the temperature at which the fish were kept. Samples were kept at 5°C overnight, the serum fraction was then isolated (3000G, 10 min), and collected by careful aspiration. The sera were kept at −80 °C until further processing.

The sampling of the injection site (or equivalent in non-injected fish), spleen, anterior kidney, heart, liver was performed 3 and 10 weeks (300 and 1020 degree days) post vaccination (10 fish/group) as previously described [20]. After sampling, each tissue was placed in 10% buffered formalin (4% formaldehyde, CellPath, UK). After 24 h, the formalin was replaced with 70% ethanol and kept at 4 °C until further processing.

### 2.6. ELISA

All sampled sera (10 fish prior to vaccinations, 10 fish/group/time point following vaccinations) were analyzed by ELISA. Unless otherwise stated, all washing steps consist three consecutive washes in a washing buffer (PBS + 0.1% Tween 20, pH 7.2). *Aeromonas salmonicida* (strain 040617-1/1A) for coating was inoculated onto a 5% blood agar plate (20 °C, 48 h) from cryoculture, and the identity of the colonies was then verified using a MONO-as agglutination kit (Bionor Laboratories A/S, Skien, Norway). Single colonies were transferred to a heat-sterilized heart-infusion broth and were incubated overnight (20 °C, constant stirring). Prior to inoculation, 100 µL of the growth medium was plated onto 5% blood agar to validate sterility. Finally, the overnight culture was washed in Millipore water and sonicated. The protein content was determined using a Pierce™ BCA Protein Assay Kit (ThermoFischer Scientific). Microtiter plates (Nunc-Immuno™ F96 MicroWell™ Maxisorp™, Thermo Scientific) were coated with 100 µL of 5 µg/mL sonicated *A. salmonicida* in a carbonate-bicarbonate coating buffer (Sigma-Aldrich) for 2 h at RT, washed and then blocked in 200 µL of blocking buffer (PBS + 5% Bovine serum albumin (BSA) + 0.1% Tween 20) for 1 h at RT. The plates were then washed again, sealed with plate seals and kept at −20 °C until needed. Each collected serum was analyzed by triplicate plating of a six-step, five-fold dilution series starting with a 1/25 dilution. All dilutions were made in a sample diluent (PBS + 1% BSA + 1% Tween 20, pH 7.2). In addition to experimental samples, two negative (1 and 2) and two positive controls (1 and 2) were included on each plate, both of which were plated in triplicate. The assay background absorbance was assessed by the inclusion of the sextuplicate plating of sample diluent alone. The volume of each sample and control added was 100 µL, and subsequently each plate was sealed and incubated for 2 h at RT. The plates were then washed and 100 µL of mouse-anti-salmonid Ig (AbD serotec, diluted 1/500) was added to each well and left to incubate (1 h at RT, 150 rpm orbital shaking). Following another wash, 100 µL of horseradish peroxidase-conjugated rabbit-anti-mouse IgG (AbD serotec, diluted 1/500) was added. After incubation (1 h at RT, 150 rpm orbital shaking), the plates were washed and 100 µL of tetramethylbenzidine substrate (TMB Sensitive, AbD serotec) was added to each well. The plates were placed on 150 rpm orbital shaking at RT and the reaction was followed closely. The reaction was stopped after 8 min by adding 100 µL of 1 M HCl, unless the reaction warranted stopping at an earlier time point. Finally, optical density (OD) values at 450 nm were measured using an Epoch™ Microplate Spectrophotometer (BioTek Instruments, Inc.).

The mean plate background OD value was subtracted from each measured sample and the control OD value. Subsequently, mean OD values for all sample and control triplicates were calculated. In order to enable comparison between different microtiter plates, interplate correction was performed. The global mean OD value measured for positive control 1 across all plates (1.93 ± 0.03, mean ± standard deviation (SD)) was divided by the plate-specific mean absorbance of positive control 1 for each plate to generate plate-specific correction factors. Mean OD values for samples and controls on each plate were then multiplied by their plate specific correction factor. For each sample, a titer was calculated based on the serial dilution. The corrected OD values for each step in the dilution series were plotted against their respective dilutions on a semilogarithmic curve with a log_10_ x-axis. Linear regression was performed on the data points located on the downward linear phase of the resulting graph, and the regression parameters were then used to calculate the point of intersection between the trajectory of the linear phase and the x-axis. This calculated dilution fold was given as the titer for that sample. A compression ratio was calculated for each plate by dividing the mean OD value of positive control 2 by that of positive control 1 to compare the dynamic capacity of the assay between plates. The mean and SD of the compression ratios was 1.61 ± 0.12. Furthermore, the coefficient of variation (CV) was calculated for each triplicate sample included in the assay, using the SD and mean of each triplicate [30]:CV = (SD/mean) × 100%(1)

The mean CV across all dilutions of all samples (1131 individual CV’s) was 8.7%.

### 2.7. Semi-Quantitative Immunohistochemical Analysis (IHC)

All tissue samples were processed through step-wise dehydration, paraffin-embedding, and sectioning (4 µm) at the Veterinary Pathology Lab, Department of Veterinary and Animal Sciences, University of Copenhagen. After sectioning, slides were dried overnight at 37 °C and stored at 4 °C until needed. From each of the two sampling time points (3 and 10 weeks post vaccination), separate sections from each of five fish per experimental group was stained for MHCII, CD8, and immunoglobulins M and T (IgM, IgT). As all primary antibodies were monoclonal hybridoma culture supernatants, an anti-*Ichthyophthirius multifiliis* hybridoma culture supernatant (kindly provided by Louise Von Gersdorff Jørgensen) was used as a negative supernatant control. For each round of processing, this negative control was used in the same dilution as the primary antibody used. In addition to this, a negative control with no primary antibody (TBS + 2% BSA) was included as well. Given the nature of each antigen and the fact that each slide included a number of tissues, these served as internal positive controls [31]. Tissue sections were processed using an UltraVision™ Quanto Detection System kit (ThermoFischer Scientific, Fremont, USA), adhering to the manufacturer’s instructions, however, with slight modifications. Tissue sections were deparaffinised in Histo-Clear II (National Diagnostics, Nottingham, England) and rehydrated through successive changes of 99%, 96%, and 70% ethanol and finally TBS. Due to the masking of antigens as a result of formalin fixation, an antigen retrieval protocol was established. A heat-induced epitope retrieval (HIER) procedure was chosen and adjusted, as suggested by Shi et al. [32]. Based on a 0.01M Tris-HCl buffer [33], a checkerboard setup was set up to cross-test pH 1, pH 6, or pH 10 with the following HIER conditions: 90 °C for 10 min, 100 °C for 10 min, or 100 °C for 20 min. The optimized conditions are shown in Table 2.

For each situation, slides were kept in the HIER solution without further heat treatment for 15 min after the treatment had ended. The optimal set of conditions was chosen for each antigen, and they are indicated in Table 2. Following HIER treatment, the slides were washed in TBS (3 min), after which they were incubated with a Hydrogen Peroxide Block reagent (10 min, kit reagent). The slides were then washed in tap water (5 min), followed by incubation with Ultra V Block (10 min, kit reagent). The blocking solution was then gently tapped off, tissues were circled using a Dako Pen (Dako, Denmark), and the slide was incubated with primary antibody overnight at 4 °C. Dilutions and references are shown in Table 3.

Following the overnight incubation, slides were gently rinsed in TBS from a spray bottle, washed in TBS (5 min) and then incubated with Primary Antibody Amplifier (10 min, kit reagent). Subsequently, slides were washed in TBS (10 min) before incubation with horseradish peroxidase (HRP) Polymer (kit reagent). This was followed by additional washes in TBS and deionized water (5 min each), and incubation with 3-amino-9-ethylcarbazole (30 min, Sigma-Aldrich, St. Louis, USA). Finally, the slides were washed in deionized water (5 min), counterstained in Mayers hematoxylin (45 s, Sigma-Aldrich), washed in deionized water, and mounted (Aquatex, Merck) with coverslips. Slides were examined at 400× magnification, and positive stains were counted for each tissue section. In addition to this, particular staining patterns, such as clustering, were noted. For illustration purposes, images were captured using the Leica Application Suite software with image smoothing disabled. Images were subsequently assembled using Adobe Illustrator CC. No image processing was performed following capture.

### 2.8. In Vitro Restimulation of Anterior Kidney Leukocytes

Fifty days post vaccination (750 degree days), a subset of fish from each experimental group were transported to the experimental facilities at the University of Copenhagen. During a three-day period 62–64 days (930–960 degree days) post vaccination, anterior kidney leukocytes were obtained from three fish per group. The fish were euthanized in 200 mg/L MS-222. The anterior kidney was aseptically removed and placed in a petri dish with an ice-chilled culture medium. The culture medium was an L15 medium (Gibco, USA) supplemented with 10 U/mL heparin (LEO Pharma, Denmark), 0.1% fetal bovine serum (Gibco, USA) and 100 U/mL penicillin, 100 µg/mL streptomycin, 0.25 µg/mL amphotericin B (Gibco, USA). Each anterior kidney was then passed through a 100 µm nylon cell strainer (Falcon, USA) into 5 mL ice-chilled culture medium and centrifuged (1000G, 10 min, 4 °C), after which the supernatant was aspirated and discarded. The pellet was washed twice and finally resuspended in 5 mL ice-chilled culture medium. The cell suspension was then transferred to a new centrifuge tube and 3 mL 34% and 3 mL 51% Percoll gradient was carefully layered below the cell suspension, followed by centrifugation (400G, 30 min, 4 °C). The cell layer located at the gradient interface was then carefully aspirated by Pasteur-pipette, then washed twice and resuspended in 5 mL ice-chilled culture medium. A 300 µL subsample of each cell suspension was used for trypan blue exclusion staining (Gibco). Counts of cells were made on FastRead 102 counting slides (Immune Systems, Devon, UK), including a live/dead count. Cells displaying erythrocyte morphology were not counted. The cell concentration was then adjusted to 1 × 10^6^ live cells/mL. Subsequently, 1 mL of cell suspension per fish were added to three separate wells on 24-well plates (Thermo Fischer Scientific, USA) and incubated at 15 °C for 24 h. The outermost wells of each plate were filled with water to limit evaporation from culture wells. Following incubation, formalin inactivated *A. salmonicida* (040617-1/1A) was added to two of the three wells to a final concentration of 2 × 10^6^, for a final ratio of 1:2 between cells and bacteria in two wells and an unstimulated well as a control. The plates were incubated at 15 °C. After 6 h, cells from one restimulated well were harvested, and after 12 h incubation, cells from the second restimulated well and the control well were harvested as well. Cells were harvested using lysis buffer (Gene Elute kit, Sigma-Aldrich), and the lysates were stored at −80 °C until further processing.

### 2.9. Isolation of RNA and cDNA Synthesis

Total RNA was extracted from the lysed, restimulated anterior kidney leukocytes using a MagMAX™-96 Total RNA Isolation Kit (Life Technologies, USA) in an automated setup on a MagMax™ Express Magnetic Particle Processor (Applied Biosystems, USA) according to the manufacturer’s instructions. Subsequently, the samples were tested for quantity and purity on a NanoDrop™ 2000 (Thermo Scientific) and were stored at −80 °C until further processing. cDNA synthesis was carried out using the TaqMan^®^ Reverse Transcription Reagents kit (Applied Biosystems), using DNase/RNase-free UltraPure™ Distilled Water (Invitrogen, Denmark). An Oligo d(T)_16_-primer included in the kit was used and the cDNA synthesis itself was carried out in µLtraAmp PCR plates (Sorenson BioScience, Inc., Salt Lake City, USA) with MicroAMP^®^ cap-strips on a T100™ Thermal Cycler (Bio-Rad, Copenhagen, Denmark) 25 °C, 10 min -> 37 °C, 60 min -> 95 °C, 5 min).

### 2.10. Real-time Quantitative Polymerase Chain Reaction (RT-qPCR)

For each RT-qPCR reaction, 2.5 µL cDNA was mixed with 2.75 µL UltraPure™ Distilled Water, 6.25 µL Brilliant III Ultra-Fast SYBR^®^ QPCR MasterMix (Agilent Technologies, USA) and a 1 µL mix of forward primer (10 µM), reverse primer (10 µM) and probe (5 µM) in AriaMx 96-well plates with optical caps (Agilent Technologies). The reactions were performed on an AriaMx Real-Time PCR System (Agilent Technologies), utilizing a 95 °C “hot start” for 3 min, followed by 40 amplification cycles (95 °C, 5 sec -> 60 °C, 10 sec). Cq-values were extracted using AriaMx Software (Agilent Technologies) and processed according to the 2^– ΔΔCt^ method by Livak and Schmittgen [37]. Elongation factor 1α (ELF1α) was used as a reference housekeeping gene [38]. For details on primers, probes and GenBank Accession numbers, see Appendix A.

### 2.11. Statistical Analyses

Unless otherwise mentioned, all statistical analyses were performed using GraphPad Prism^®^ (San Diego, USA). A 95% confidence level was employed for all statistical analyses. Consequently, for all comparative analyses, *P*-values < 0.05 resulted in the rejection of the null-hypothesis that no difference exists between the examined groups. Antibody titers were examined for Gaussian distribution using the D’Agostino and Pearson omnibus normality test, and finally compared using the Kruskal–Wallis test followed by a Dunn’s post test to identify statistically significant differences between groups. For the comparison of ΔCq-values, a two-tailed, homoscedastic Student’s *t*-test was performed using Microsoft Excel. In addition to the aforementioned *P*-value requirement, for RT-qPCR data, a minimum fold-change of 2 was used as a threshold for considering changes in transcription levels valid.

## 3. Results

### 3.1. ELISA

The development of *Aeromonas salmonicida*-specific Ig in sera from each experimental group sampled at three, six, and ten weeks post vaccination were detected, as detailed in the Materials and Methods section. Mean titration curves for each group at each of the three time points are shown in Appendix A. Calculated titers are shown in Figure 1. At three weeks post vaccination, statistically significantly increased antibody titers relative to the naïve control group were found in the bacterin group (*P* < 0.01) and the FIA group (*P* < 0.0001). Six weeks post vaccination, the bacterin (*P* < 0.05), CpG (*P* < 0.0001), CAF01 (*P* < 0.05), and FIA group (*P* < 0.0001) titers were all statistically significantly elevated relative to the naïve control group values. At the final sampling time point ten weeks post vaccination, statistically significant increases in antibody titer relative to naïve control values were seen for the bacterin (*P* < 0.05), CpG (*P* < 0.01), and FIA group (*P* < 0.001). No significant difference from the naïve control group was seen for the CAF01 group at this time point.

### 3.2. Semi-Quantitative Immunohistochemical Analysis

In order to evaluate immune responses on a cellular level, IHC staining for key immune markers was performed for each experimental group. Representative stains are shown in Figure 2. As a general note, the necessary heat-induced epitope retrieval (HIER) procedures proved to cause some damage to injection site sections. Typically, the epidermis and scales were affected, as well as the peritoneum. These damages were taken into consideration when evaluating sections.

### 3.3. MHCII-Specific Staining

Three weeks post vaccination, low numbers, if not an absence, of MHCII^+^ cells was observed in sections from the naïve control, TBS, bacterin, and CpG-formulation group. Observations of MHCII^+^ cells were made in some anterior kidney sections across these four groups, with very few or no positive cells seen in the spleen, heart, injection site, or liver. For the CAF01- and FIA-formulations, however, a stable presence of MHCII^+^ cells was observed in the spleen, anterior kidney, and liver (Figure 2A). In sampled injection sites, MHCII^+^ cells were seen in the epidermis, typically near mucus cells. A general tendency of immunoreaction in cells surrounding vessels in the liver was noted. At ten weeks post vaccination, the presence of MHCII^+^ cells was generally more pronounced when compared with that observed at three weeks post vaccination. Naïve control and TBS group fish now showed stable but low numbers of MHCII^+^ cells in the spleen, anterior kidney (Figure 2B) and liver, while only one fish in the TBS group had MHCII^+^ cells in the heart. MHCII^+^ cells in the spleen and anterior kidney showed no clear tendency towards clustering in these groups. In the bacterin group, as well as the three vaccine groups, a noticeable increased presence of MHCII^+^ cells was observed in the anterior kidney, often appearing in clusters. Furthermore, an increased presence of MHCII^+^ cells in the white pulp of spleen sections was observed for these four groups. For the FIA group, MHCII^+^ cells in the spleen were typically found to cluster together, while the other groups showed no sign of similar clustering. In each of the three vaccine groups, MHCII^+^ cells were seen in heart sections. Finally, MHCII^+^ cells were observed in the epidermis of the injection site sections for the bacterin, CpG, and FIA groups. In the CAF01 group individuals, slightly increased numbers of MHCII^+^ cells were seen in these sections.

### 3.4. CD8 Specific Staining

For the CD8 specific staining, some level of background staining was observed. This was taken into consideration when examining the sections. With the exception of one individual in the naïve control group, fish from this and the TBS groups displayed very few, if any, CD8^+^ stains in the spleen, heart, injection site, and liver at three weeks post vaccination. In the TBS group, low numbers of CD8^+^ cells were observed in the anterior kidney (Figure 2C). In the bacterin group and the three vaccine groups, CD8^+^ cells were observed in epidermis, and a strong, uniform, and general presence was seen in the spleen, anterior kidney, and liver sections. CD8^+^ cells in the spleen and anterior kidney appeared spread throughout each section. Furthermore, CD8^+^ cells were observed in heart sections of bacterin, CpG, CAF01, and FIA group fish, with a pronounced presence in heart sections from CpG group fish. At ten weeks post vaccination, a drastic reduction in the presence of CD8^+^ cells was seen for bacterin and CpG group fish in all sampled tissues when compared to three weeks post vaccination. Some reduction was also seen for the CAF01 group, particularly in the spleen and liver sections, while the FIA group maintained elevated levels of stained cells in both the spleen and anterior kidney.

### 3.5. IgM Specific Staining

With very few scattered exceptions, no IgM^+^ cells were found in the heart, injection site, or liver sections in any of the experimental groups, at any of the two time points. At three weeks post vaccination, a steady presence of IgM^+^ cells was found in anterior kidney sections of all six experimental groups. Low numbers of IgM^+^ cells were observed in the spleens of fish in the naïve control, TBS, CpG, and CAF01 group, although the latter had one individual with an increased number of IgM^+^ cells. Across five examined fish, only a single positive cell was found in the spleen sections of the FIA group. Ten weeks post vaccination, IgM^+^ cells were primarily found in the anterior kidney. In spleen sections from the naïve control, TBS, CAF01, and FIA groups only isolated findings of very low numbers of IgM^+^ cells were seen. Only the bacterin and CpG groups showed a steady presence of low numbers of IgM^+^ cells in spleen sections, widely found in white pulp and in clusters (Figure 2D). In the anterior kidney sections, a steady presence of IgM^+^ cells was found in all groups, although noticeably few stained cells were observed in the CAF01 group.

### 3.6. IgT Specific Staining

A relatively large variation in IgT^+^ cells was found between individuals across the various tissue sections. In several cases, individuals with strongly elevated numbers of IgT^+^ cells showed elevated MHCII^+^ cell counts, as well. IgT^+^ cells were present in the spleen, anterior kidney, heart, and liver of all groups. In the liver sections, IgT^+^ cells were typically observed both in tissue surrounding vessels, as well as within the vessels themselves (See Figure 2E). In addition to those, isolated findings of IgT^+^ cells were reported from injection site sections of the naïve control, TBS, bacterin, and CpG groups, where they were observed in the dermis near mucus cells. None were observed in the CAF01 and FIA groups, but otherwise no discernible differences were found between groups at three weeks post vaccination. Observations from the heart and liver sections in general, as well as from clotted blood found on one slide, suggest that a high number of IgT^+^ cells were in circulation three weeks post vaccination. A decrease in general numbers of IgT^+^ cells was seen between three and ten weeks post vaccination, and variation between individuals was still found, in particular for the CAF01 and FIA groups. With the exception of two individuals, one in the naïve control group and one in the CpG group, no IgT^+^ cells were observed in injection site sections. Across all groups, a reduced number of individuals were found to have IgT^+^ cells in the heart sections. For the bacterin group, as well as the three vaccination groups, individual IgT^+^ cells were observed in direct contact with melanized cells in the individual spleen or anterior kidney sections (Figure 2F).

### 3.7. RT-qPCR

In order to address whether or not each experimental vaccine led to the development of an immune profile comparable to one of the known T_H_ cell profiles, in vitro restimulation of anterior kidney leukocytes from each experimental group was performed. The responses of the leukocytes were then investigated on a transcriptional level, focusing on immune-relevant cytokines and transcription factors. The transcription of key cell markers was also examined to add to the observations made using IHC. Finally, the transcription of members of the complement system was assessed as well. Results are shown in Figure 3. The transcription levels are analyzed in three stages, each utilizing a different reference group. No transcription of MHCII or IL-17A/F2 was found in any of the samples. A comprehensive overview of the findings is included in the supplementary material (Appendix A). Finally, the relative gene expression levels of each gene are illustrated in Appendix A.

### 3.8. TBS Group as Reference Group

To address the effects of each immunization, transcription levels from each of the immunized groups were compared to those of the TBS group. Comparing values for 6 h stimulation, 12 h stimulation, and no stimulation for each of the immunized groups to their respective TBS group counterparts, the effects of each immunizing injection were examined (See Figure 3). For the bacterin group, a statistically significant decrease in the transcription of C5 was seen after 6 h restimulation, while increased FoxP3A, IFN-ϒ, IL-10, and IL-17C1 transcription levels were seen following 12 h of restimulation. The CpG group showed a significant increase in IL-10, IL-17C1, and IL-17C2 transcripts, also after 12 h of restimulation. For the CAF01 group, significantly upregulated IFN-ϒ, IL-17C1, and IL-17C2 transcription was observed, again after 12 h of restimulation. However, in addition to this, a statistically significant increase in the expression of CD8, IgM, and IgT was found for the unstimulated CAF01 group leukocytes. No other statistically significant differences were found for unstimulated cells at this stage of analysis. Finally, significant increases in the transcription of IL-10 and IL-17C1 for the FIA group of leukocytes after 12 h of restimulation.

### 3.9. Bacterin Group as Reference Group

To address the effects of each adjuvant included in the experimental vaccine formulations, the transcription levels of these three groups were compared to those of the bacterin group (See Figure 3). When compared to the bacterin group, GATA3 and IFN-ϒ were significantly upregulated in the CpG group in unstimulated cells and after 6 h of restimulation, respectively. Meanwhile the CAF01 group showed a significantly increased transcription of IgM in unstimulated cells, as well as a a significantly decreased transcription of C3 after 12 h of restimulation. In the FIA group, significant upregulations of both C5 and GATA3 were found in unstimulated leukocytes.

### 3.10. Unstimulated Leukocytes as a Reference Group

Comparing the transcription levels of 12 h restimulated leukocytes of each group to their respective unstimulated controls, revealed some statistically significant differences in transcription levels (see Appendix A). In naïve control group leukocytes, restimulation resulted in the significant downregulation of FoxP3A, IgM, and RORϒ, while significant upregulation was seen for IL-17C1 and TNFα. For TBS group cells, only IL-17C1 was significantly upregulated, while a significant upregulation of GATA3 was found after 12 h of restimulation in the bacterin group. The CpG group revealed a significantly increased transcription of IL-6, IL-12, IL-17C1, and TNFα following restimulation, while only the upregulation of IL-17C1 and IL-17C2 was statistically significant in the CAF01 group. Finally, a 12 h restimulation of the FIA group leukocytes resulted in a statistically significant decrease in both CD8 and TGFβ transcripts and significant increases in IL-17C1, IL-17C2, and TNFα transcripts, when compared to unstimulated leukocytes.

## 4. Discussion

An emphasis on antigen-specific antibody titers has previously been expressed, directly associating this immune mechanism with fish vaccine efficacy and protective immunity [3,39,40,41,42]. Indeed, in the present study, consistently increased titers were demonstrated early after vaccination for the bacterin and FIA groups, when compared to the naïve control group. For the CpG and CAF01 groups, however, the development of significantly elevated Ig titers was slower. Once elevated, the CpG and FIA groups remained significantly elevated throughout the experimental period. The CAF01 group, however, displayed a low, transient, but statistically significant elevation compared to those of the naïve control group only at six weeks post vaccination. This is an interesting dynamic, given that rainbow trout injected with mineral oil adjuvanted vaccines have been demonstrated to maintain elevated antibody levels for at least 14 months [17]. The CAF01 adjuvant system has previously been shown to be a potent inducer of IgG1 in a murine model following triple immunizations with ovalbumin, matching the performance of an immunization scheme using a primary immunization formulated with complete Freund’s adjuvant, followed by booster immunizations formulated with FIA [25]. The same study demonstrated a strong induction of IgG2 in three different disease models. Currently, we do not have access to antibodies that enable distinction between isotypes in trout antibody responses by ELISA. Nonetheless, the magnitude of the antibody responses seen for the CAF01 group individuals appears lower than those of the other three immunized groups. As we have no previous experience with the CAF01 adjuvant system in fish, any differences between results from murine and teleost models could likely be attributed to interspecies differences in immune mechanics and kinetics, as well as the absence of booster immunizations in the present experimental setup. Differences in Ig kinetics are of interest, given the abovementioned emphasis on circulating antibodies following vaccination. In support of this, the apparent depletion of specific antibodies during experimental, waterborne infection of rainbow trout [41] seems to provide a mechanistic explanation of the importance of specific antibodies in protective immunity. However, based on statistically significant experimental data, this line of reasoning does not take any cell-mediated mechanisms into account. A more prominent role of cell-mediated immune responses could thus account for the protection induced by the CAF01-adjuvanted experimental vaccine. The statistically significant increases in specific antibody levels in rainbow trout injected with un-adjuvanted *A. salmonicida* bacterin is interesting, as this was not seen following the bath vaccination of the trout, using the same bacterial isolate [43]. A similar induction following injection has nevertheless been demonstrated by Veenstra et al. [44].

While the apparent increase in the presence of MHCII^+^ cells in the naïve control, TBS, bacterin, and CpG-ODN groups may reflect a development or maturation of the immune cell repertoire of these groups, the results from the emulsified vaccine groups indicate an earlier involvement of this cell type for the CAF01 and FIA groups. All immunized groups show increased numbers of MHCII^+^ cells in the anterior kidney, as well as in the white pulp of the spleen sections at ten weeks post vaccination. However, the noticeable presence of MHCII, a cell-marker associated with antigen-presenting cells (APCs), such as macrophages and dendritic cells [45,46], in the anterior kidneys and spleens of CAF01 and FIA group individuals three weeks post vaccination, is indicative of a depot effect causing the prolonged processing of antigens and delayed clearance of the antigenic content, specific to these two formulations. This supports the observed differences in the activation of the MHCII^+^ cells mentioned earlier, and is itself further supported by the fact that the noted high presence of CD8^+^ T-cells in immunized groups was only sustained by the CAF01 and FIA groups, and only for the spleen and anterior kidney. As the anterior kidney serves as lymphoid tissue in teleosts, and given the filtering function of the kidney and spleen [47], these organs are crucial in immune functions. Therefore, these observations are in line with the expected slower, continual release of an antigen from particulate vaccine formulations, underlining a specific mode of action for both the CAF01 and the FIA formulation. This also indicates that although Ig titers of the CAF01 group, as opposed to the FIA group, are no longer significantly higher than those of the naïve control group after ten weeks, the prolonged stimulation of adaptive immune mechanisms is still occurring. Along with the significant increase in CD8 transcripts observed between the unstimulated TBS and CAF01 groups, this further supports the suggestion of a more prominent, baseline role of cell mediated responses in CAF01-induced immunity, as mentioned previously.

While completely absent from naïve control and TBS group individuals, the observation of MHCII^+^ cells in the epidermis of immunized groups, although few in numbers, suggests an increased state of mucosal immune preparedness in immunized fish. While both secreted IgT and IgT^+^ B-cells have been identified as important components of mucosal immunity [48,49,50], IgT^+^ cells were only observed in low numbers in epidermis, and primarily at three weeks post vaccination. Although the HIER treatment caused mechanical damage to injection site sections, and the epidermis in particular, the findings must be regarded as highly infrequent. The absence of IgT^+^ cells in the epidermis of CAF01 and FIA group individuals must nevertheless be interpreted with caution. 

Given that no IgM^+^ cells were observed in epidermis, IgT^+^ cells were still technically the more prevalent B cell subset, which is consistent with previous findings [48], however, only by a small margin. Outside of the epidermis, the long-term presence of IgT^+^ cells in the anterior kidney and spleen sections of immunized fish observed in apparent direct cell–cell contact with presumed melanomacrophages, indicates antigen presentation between APCs and recipients, in this case B-cells, in these tissues.

IgM^+^ cells, unlike IgT^+^ cells, appeared highly focused regarding their tissue distribution. Found almost exclusively in the spleen and anterior kidney, at three weeks post vaccination, only the bacterin and CpG group show a sustained, albeit reduced presence in the spleen at ten weeks post vaccination. For the remaining groups, IgM^+^ cells are only really found in anterior kidney at this point. This is in good accordance with previous studies, stating that antibody-secreting B cells were initially found in the circulation, spleen, and anterior kidney of a rainbow trout after immunization, whereas after approximately 10 weeks their numbers decreased in the circulation and spleen, but not in the anterior kidney [51]. This pattern has also been associated with the transition from the resting B cell, through the plasmablast stage, both of which occur in spleen and circulation, to finally reach the state of the plasma cell in the anterior kidney of rainbow trout [52,53]. In this study, the particulate vaccine formulations (CAF01 and FIA groups) appear to undergo a similar shift in distribution of IgM^+^ cells during the experimental period. A noteworthy finding in that context, is that while the particulate formulations retained a strong presence of CD8^+^ T-cells, IgM^+^ cells appear in relatively low numbers in these groups at both time points, with particularly low numbers found in the CAF01 group. With regards to IgM^+^ cells, the non-particulate formulation and the bacterin alone sustain higher numbers over time.

The transcription data from the in vitro restimulation assay proved more complex than the textbook framework of the traditional T_H_ profiles. A number of factors should be considered when examining this dataset. First of all, practical limitations on experimental animal sample sizes proved to be an important limiting factor. In terms of the antigenic content used in this study, a whole-cell bacterin introduces a high number of potential pathogen-associated molecular patterns, which in itself must be expected to lead to a complex reaction pattern. This should be avoided by using an immunologically inert subunit antigen in future studies. As such, the experimental setup and the expected dissimilar responses from subsets within the large number of individual leukocytes involved must be considered as confounding factors. Nevertheless, the transcription data identify shared, as well as group-specific patterns, further indicating distinct differences among the immunized groups. Using TBS-injected fish as the reference to identify transcription patterns associated with immunizing injections, the increased transcription of IFN-ϒ, IL-17C1 and IL-10 was seen across the immunized groups. The expression patterns observed for IFN-ϒ and IL-10 were found to be highly similar, except for the fact that the transcription level of the IFN-ϒ was approximately three times higher than that of IL-10 (see Appendix A). As IFN-ϒ, IL-17C1, and IL-10 are typically associated with T_H1_, T_H17_, and both T_reg_ and T_H2_ profiles, respectively, the injection of whole-cell *A. salmonicida* bacterin appears to cause a complex immune response by itself. This seems reasonable given its success in at least some unadjuvanted bath vaccine strategies [43].

Besides these common expression patterns, bacterin-injected fish showed a statistically significant upregulation of FoxP3A, a transcription factor associated with the mammalian, as well as the teleost regulatory T cell (T_reg_) subset [54]. When using the bacterin group as a reference, we observe that the CpG group demonstrated a significant increase in IFN-ϒ transcription, a transcription pattern that was not seen for either of the other two adjuvants, but is in line with previous studies in Atlantic salmon, where an increased transcription of IFN-ϒ was seen from anterior kidney cells following the injection of CpG ODNs [24]. This pattern of IFN-ϒ expression, along with the statistically significant increase in the transcription of both IL-12 and TNF-α observed when comparing transcription in unstimulated leukocytes to that of stimulated leukocytes from the same experimental group, suggest that the CpG group leukocytes are approaching a T_H1_ profile [55].

In murine and human studies, the CAF01 adjuvant system has been shown to induce a mixed T_H1_/ T_H17_ profile primarily based on the secretion of IFN-ϒ, TNF-α and increased levels of IgG2 [25,56,57]. In the present study, non-stimulated leukocytes from the CAF01 group showed a significantly upregulated transcription of CD8, IgM, and IgT, while a 12 h restimulation induced a significant transcription of IFN-ϒ, as well as two isoforms of IL-17, when compared leukocytes from TBS-injected fish. While some similarities are observed between the mammalian and trout models, the influence of the whole-cell antigen should be considered. The increased expression of CD8 correlates well with the observation of increased numbers of CD8^+^-cells in this group. The relative increase in IgM^+^-cells is not reflected in the serum Ig titers or in the immunohistochemical observations. Since both IFN-ϒ and IL-17 expression are seen across the immunized groups, only CD8 and immunoglobulin transcription stands alone, and a placement within a certain T_H_ profile becomes questionable. However, the relatively low numbers of IgM^+^ cells, the increased transcription of IgM and IgT and the transient nature of the Ig titers must be considered as indicative of a particularly complex humoral response in this group, warranting further investigation in order to be fully clarified.

## 5. Conclusions

While vaccines based on identical antigenic content, but different adjuvants, were found to be similar in some aspects, differences were found as well. In particular, the difference between the two particulate vaccine formulations in terms of antibody titers and the suggested resultant increased reliance on cell-mediated responses suggests that while an efficient humoral response is most likely important, important nuances can be exploited. While significant reactions were observed in fish injected with unadjuvanted bacterin, these are not expected to last throughout a rainbow trout production cycle in light of earlier studies [19], reinforcing the emphasis on adjuvants.

In conclusion, the results from the present study demonstrate a significant extension of the array of immune mechanisms that can be successfully targeted to achieve protective immunity against furunculosis infections in rainbow trout, enabling the development of future injection vaccines with improved, more acceptable safety profiles.

## Figures and Tables

**Figure 1 vaccines-08-00103-f001:**
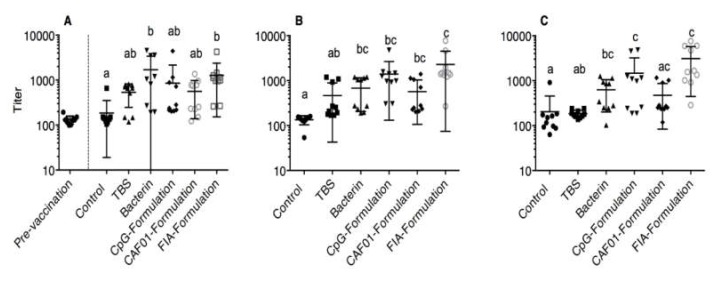
Scatter Dot-Plots of mean ELISA titers ±SD for each experimental group. For details on titer calculation, see the Materials and Methods section. (**A**) Three weeks post vaccination: the pre-vaccination values are shown as a point of reference and are not included in the statistical analysis. (**B**) Six weeks post vaccination. (**C**) Ten weeks post vaccination. Group titers at each time point were compared by Kruskal–Wallis, followed by Dunn’s post test. Statistically significant differences are denoted using lower case letters. Identical letters indicate *P* > 0.05, different letters indicate *P* < 0.05.

**Figure 2 vaccines-08-00103-f002:**
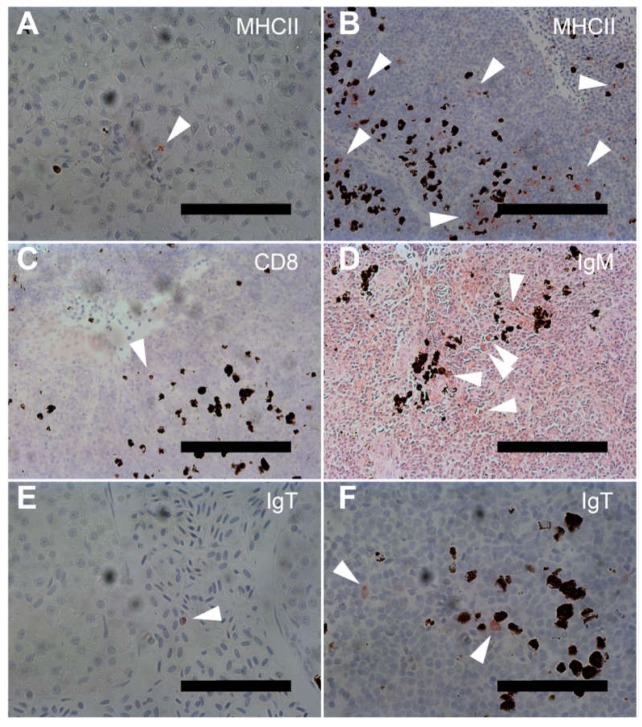
Immunohistochemical staining of tissue sections. Staining was performed as described in the Materials and Methods section. Arrowheads are placed to aid identification of immunopositive cells. (**A**) MHCII-immunopositive cell in liver section, naïve control group, three weeks post vaccination, 400×. Scale bar (SB) = 75 μm. (**B**) MHCII-immunopositive cells in the anterior kidney section, naïve control group, ten weeks post vaccination, 200×, SB = 150 μm. (**C**) CD8-immunopositive cell in the anterior kidney section, tris-buffered saline (TBS) group, three weeks post vaccination, 200×, SB = 150 μm. (**D**) IgM-immunopositive cells in the spleen section, CpG group, ten weeks post vaccination, 200×, SB = 150 μm. (**E**) IgT-immunopositive cell in liver section, naïve control group, three weeks post vaccination, 400×, SB = 75 μm. (**F**) IgT immunopositive cells in the anterior kidney section, CAF01 group, ten weeks post vaccination, 400×, SB = 75 μm.

**Figure 3 vaccines-08-00103-f003:**
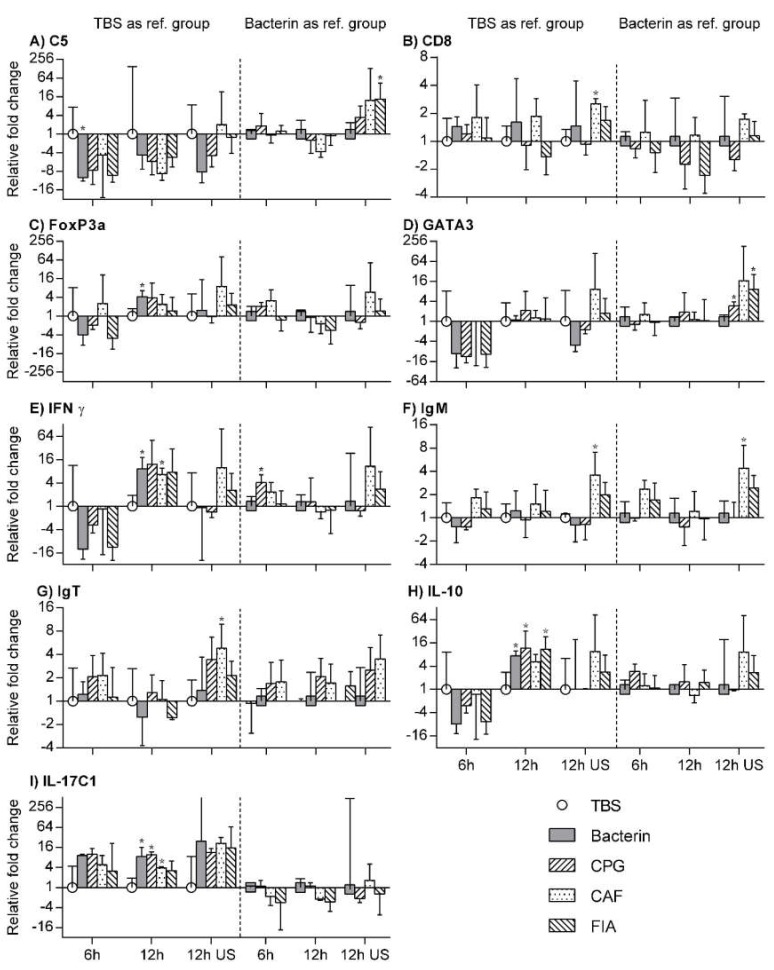
Comparative analysis (**A**–**I**) of RT-qPCR data from in vitro restimulation of isolated anterior kidney leukocytes. The x-axis denotes the duration of the restimulation (“US” denotes unstimulated cells). Each individual graph illustrates the fold change (± geometric mean) in the transcription of the target gene (identified in the upper left corner of each figure) for a given experimental situation and group relative to that of the TBS group (left panel, TBS group denoted by a white circle) and to that of the bacterin group right panel, bacterin group denoted by filled, grey square). *Asterisks denote a statistically significant difference from the reference group values.

**Table 1 vaccines-08-00103-t001:** Designation and content of experimental groups.

Group Designation:	Injection:	TBS:	Antigenic Content:	Adjuvant:
Naïve controls	-	-	-	-
TBS group	+	+	-	-
Bacterin group	+	+	1 × 10^8^ CFU	-
CpG ODN group	+	+	1 × 10^8^ CFU	0.6 nM CpG ODN in TBS
CAF01 group	+	+	1 × 10^8^ CFU	300 µg CAF01 in TBS
FIA group	+	+	1 × 10^8^ CFU	50 µL FIA in TBS

**Table 2 vaccines-08-00103-t002:** Optimized heat-induced epitope retrieval (HIER) treatment of tissue sections for each antigen.

Target Antigen(s):	Optimized HIER Conditions:
IgM	100 °C, 20 min, pH 6
MHCII, CD8, IgT	100 °C, 20 min, pH 10

**Table 3 vaccines-08-00103-t003:** Monoclonal antibodies used in the present study.

Target/Epitope	Dilution	Reference
MHCII (F1-6, beta 2 domain of Atlantic salmon MHCII beta chain)	1:750	[34,35]
CD8α (F1-29, membrane distal Ig-like domain of Atlantic salmon CD8α)	1:150	[34,35]
IgM (F1-18, anti-trout IgM)	1:1800	[36]
IgT (F1-8, second constant domain of the rainbow trout IgT heavy chain)	1:300	[34]

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
