# Peer review of "Injection Vaccines Formulated with Nucleotide, Liposomal or Mineral Oil Adjuvants Induce Distinct Differences in Immunogenicity in Rainbow Trout"

_vaccines, 2020, doi:10.3390/vaccines8010103_

Round 1

Reviewer 1 Report

It is an interesting paper that demonstrates a good work done.

However, I have two recommendations:
1. Material and methods could be abbreviated since some parts are supported by previous studies by the same authors.

2. Results from the present study should not be compared studies conducted in murine and human models, since it is already complex to compare studies between rodents and other animal species... so, it should be justified very well to compare with studies developed in fishes.

Author Response

Response to reviewer 1 following peer review:

Please find the reviewers responses in quotes below, followed by our italicized responses to each point raised.

”It is an interesting paper that demonstrates a good work done.”

  • We appreciate this. Thank you.

“However, I have two recommendations:
1. Material and methods could be abbreviated since some parts are supported by previous studies by the same authors.”

  • We accept this. The section on Fish (Page 3, lines 97-101), as well as the section on Vaccine preparation (page 3, lines 103-120) has now been reduced, both with reference to previous publication. We prefer to leave the remaining sections as they are, to make sure that reproducibility is not compromised.

“2. Results from the present study should not be compared studies conducted in murine and human models, since it is already complex to compare studies between rodents and other animal species... so, it should be justified very well to compare with studies developed in fishes.”

  • However, in this case CAF01 has never been used in any species of fish prior to this study. Our assumptions for using this adjuvant formulation is therefore based solely on the performance and reactions observed in murine models. This is why we chose to include these in the discussion.

Further changes based on submitted reviewer PDF:

  • In line 105, “dr.” has been changed to “Dr.”.
  • In line 162, a semicolon was added.
  • In line 253, we are not aware what the request is. It might be meant as a remark on the relatively low number of fish sampled. We are aware of this, as reflected in the discussion.
  • In lines 331 and 491 the reference to “Ig” refers to the antibody used in the ELISA setup. As the primary antibody is anti-salmonid Ig (as described in Materials and Methods), it does not distinguish between isotypes. This is reflected in the discussion on CAF01 antibody dynamics.
  • Regarding the referencing of human and murine studies in line 597, this has already been addressed above.
  • Regarding the lack of bold type in references 8 and 30 (lines 665 and 730), we have tried to make our reference software cooperate.
  • The request to use WebCite is much appreciated. However, as of today February 17th, the cite is not accepting further requests for archiving. We have used WayBack Machine as an alternative, with all links provided.

Reviewer 2 Report

The Manuscript is quite extensive and complex. It is hard work to read the difficult contents and it would have been better to publish the results at least in two separate parts.

The writing is quite complicated and could be done in a more simple understandable way.

The results should consequently be presented in one tense.

The text should be looked through carefully for errors like:

Title and text passages: As the term "Rainbow trout" usually means more than one fish, the correct term should be "rainbow trouts" or "the rainbow trout".

line 72: ... with regard to ...

line 75. ... included: ...

line 78: ... distribution ...

line 79: ... was investigated ...

line 81: ... anterior kidneys from each ...

line 129: ... volume forms ...

line 207: ... embedding (in what??) ... and sectioning (to how many µm?) ...

line 211: ... fish per experimental ...

line 214/15: ... process of samples (?) ...

lione 233. ... gently taped off, ...

and so on.

Author Response

Response to reviewer 2 following peer review:

Please find the reviewers responses in quotes below, followed by our italicized responses to each point raised.

“The Manuscript is quite extensive and complex. It is hard work to read the difficult contents and it would have been better to publish the results at least in two separate parts.”

  • We acknowledge that the manuscript is rather complex. Nevertheless, we have sought to be concise and detailed in the descriptions to allow full understanding and reproducibility of the presented study. We disagree, however, with the suggestion to divide the manuscript into two parts, as this would disrupt the coherence of the separate approaches when attempting to make overall conclusions.

“The writing is quite complicated and could be done in a more simple understandable way.”

  • As stated above, we have tried to present the study in a concise and reproducible manner.

“The results should consequently be presented in one tense.”

  • The results section should be in past tense only. Upon revision of this section, lines 441-443 are written in present tense. As they describe inclusion of supplementary material and a referral to the following figure, we will argue that it is appropriate.

“The text should be looked through carefully for errors like:”

“Title and text passages: As the term "Rainbow trout" usually means more than one fish, the correct term should be "rainbow trouts" or "the rainbow trout".”

  • The definite article was added in table 3. Otherwise, all mentions of rainbow trout were meant to be plural.

“line 72: ... with regard to ...”

  • We agree. This has now been corrected.

“line 75. ... included: ...”

  • We agree. This has now been corrected.

“line 78: ... distribution ...”

  • We agree and have now corrected this.

“line 79: ... was investigated ...”

  • We agree. This has now been corrected.

“line 81: ... anterior kidneys from each ...”

  • We agree. This has now been changed.

“line 129: ... volume forms ...”

  • We disagree. This is not the intended wording.

“line 207: ... embedding (in what??) ... and sectioning (to how many µm?) ...”

  • These details have been added to line 229 of the revised manuscript.

“line 211: ... fish per experimental ...”

  • We accept this, and this has been changed in what is now line 233 of the revised manuscript.

“line 214/15: ... process of samples (?) ...”

  • We agree, and have now corrected this in what is now line 237.

“lione 233. ... gently taped off, ...”

  • We disagree. The solution was indeed tapped off.

and so on.

  • ­The manuscript has now been revised, and the peer-review process has improved it.